# Adhesive Wear and Frictional Performance of Banana Fibre Reinforced Epoxy (BaFRE) Composite

**DOI:** 10.3390/polym14173700

**Published:** 2022-09-05

**Authors:** Umar Nirmal

**Affiliations:** Faculty of Engineering and Technology, Multimedia University, Jalan Ayer Keroh Lama, Melaka 75450, Malaysia; nirmal@mmu.edu.my

**Keywords:** banana fibre, dry contact, fibre orientation, BOD, Ws, friction, temperature, epoxy, RA

## Abstract

This current work is an attempt to investigate the tribological performance of banana fibre-reinforced epoxy (BaFRE) composite in dry contact conditions. The test is conducted on a wear test rig of type Block on Ring (BOR) based on ASTM G77, G137-95 standard. Different applied normal loads (5–30 N) subjected to a stainless steel counterface at different sliding speeds (1.7–3.96 m/s) and fixed sliding distance of 6.72 km were set as the experimental parameters. The test was conducted using neat epoxy (NE) as the control tests’ specimen while BaFRE composite was in anti-parallel (AP) and parallel (P) fibre strips orientation (O). The specific wear rate (Ws), friction coefficient, temperature variation and surface roughness (RA) of the NE and BaFRE composite in AP-O and P-O were investigated. The worn surface morphology of the test samples was examined under a high-resolution metallurgy microscope. The outcome of the work revealed that AP-O exhibited excellent wear performance when compared to P-O and NE. Moreover, the lowest friction coefficient of 0.0121 is achieved when AP-O is subjected to 30 N of applied load and 3.96 m/s of sliding velocity. BaFRE composite in AP-O demonstrated the lowest range of temperature variation when it was subjected to 30 N at 3.96 m/s of counterface sliding velocity. Due to the high shear resistance at the rubbing zone incurred by the AP-O test specimen and counterface, the RA values were remarkably high for the test specimen (i.e., 1.122 µm) and the counterface wear track zone (0.166 µm) as compared to the P-O and NE test samples. The predominant wear mechanism for the AP-O test specimen was plastic deformation, detached fibre, micro crack, minor fibre debonding and micro ploughing. In conclusion, there are improvements in terms of wear and friction performance of the composite when banana fibres are used as a reinforced element in epoxy resin. The improvement in Ws and friction coefficient for the AP test specimen was 29.4% and 48.6%, respectively, as compared to NE at 30 N of applied load, 6.72 km of sliding distance and 2.83 m/s of sliding speed.

## 1. Introduction

Polymer based materials have replaced metal parts in various industries such as manufacturing of air-planes and cars [1]. This is because they possess advantages such as lower density, less need for maintenance and lower cost [2]. The thermosetting polymers are the materials of choice for long-term utilization in many applications as they are infusible, insoluble, and have high-density. However, the increasing production of thermosetting polymers results in a large amount of waste materials that have caused serious waste problems [3]. Hence, the usage of renewable resources such as plants or animals based fibre-reinforced polymeric composites has received great attention since it has emerged as a potential environmentally friendly and cost-effective measure. Moreover, these natural fibres are available abundantly in many developing countries, which could easily fulfil the demand of scientists and engineers to use them effectively. Furthermore, these natural fibres offer brilliant structural and functional features such as high specific stiffness and specific strength to the fibre reinforced composites due to their unique combinations of properties [4].

In recent years, there have been concerns related to the tribological behaviour of natural fibres-reinforced polymeric composites such as jute [5], cotton [6], sugarcane [7], sisal [8], betelnut [9], bamboo [10], oil palm [11], coir [12] and kenaf [13]. The surface characteristics, volume fraction, physical properties and orientation of natural fibres play an essential part in controlling the mechanical and tribological behaviour of the composites. Previous studies stated that the orientations of the fibres in the matrix influence the wear results. It is reported that superior wear resistance was achieved when the orientation of the fibres was anti-parallel to the sliding surface [9]. In 2008, N.S.M. El-Tayeb [14] discovered that the AP-O of sugarcane fibre to the sliding direction performed better in wear resistance than the P-O subjected to the tribological behaviour of polyester composites.

In 2010, Umar Nirmal et al. [15] stated that contact conditions have a great influence in controlling the tribo performance of polymeric composites. Previous studies reported that some polymeric composites, under wet contact conditions, have better tribological performance compared to dry contact conditions [16,17].

Different types of natural fibres are used as reinforcement in polymers, and they displayed different tribology behaviours [18]. Figure 1 shows the works on the tribological performance on various types of natural fibres’ composites. From the figure, it can be observed that sisal fibre-reinforced with polyester resin shows very high friction coefficient with low specific wear rate. In this way, it could be interpreted that the wear and frictional behaviour of the composites do not have any correlation. In addition to that, bamboo/epoxy composite exhibits a moderately low friction coefficient and has a high specific wear rate. Brilliant wear performance with superior friction resistance in tribological applications is desired, such as in the use of bushes, bearing and sliding rails used in the construction industry. Further, to improve the friction performance of natural fibre/polymer composites, it was reported in [15] to operate the composites under wet contact conditions or in the addition of solid lubricants during composite fabrication.

According to this literature survey, it was noted that there is less work reported on banana fibres-reinforced epoxy composite for tribological applications. Moreover, Malaysia is a blessed country as it is located at the equatorial region which receives good annual sunlight with average rain fall. This promotes the growth of various agricultural plants. As a result of harvesting these plants, the agricultural waste is not properly utilized, which poses a health hazard when they are burnt openly in the rural areas in Malaysia. Hence, we took the effort to study the usage of banana fibres as potential reinforcements in polymeric composites for tribological applications. The test will be conducted on a wear test rig of type Block on Ring (BOR) based on ASTM G77, G137-95 standard [19]. Different applied normal loads (5–30 N) subjected to a stainless steel counterface at different sliding speeds (1.7–3.96 m/s) and fix sliding distance of 6.72 km will be used as the experimental parameters. The test will be conducted using neat epoxy (NE) as the control test specimen and BaFRE composite in anti-parallel (AP) and parallel (P) fibre strips orientation (O). The specific wear rate (Ws), friction coefficient, temperature variation and surface roughness (RA) of the NE and BaFRE composite in AP-O and P-O will be investigated. The worn surface morphology of the test samples will be examined under a high-resolution metallurgy microscope.

## 2. Methodology

### 2.1. Flow Chart of Experimental Works on Wear and Frictional Performance of BaFRE Composite

Figure 2 summarizes the necessary stages in the realization of the current work. In a nutshell, there are eight crucial stages involved. All of these stages are explained in the subsequent sections, respectively.

### 2.2. BOR Experimental Procedures

Tribological behaviour of the BaFRE composite was tested using a BOR machine subjected to adhesive dry contact conditions, as shown in Figure 3. It was manufactured in accordance to ASTM G77, G137-95 standards. In general, this test was imitated for applications such as rolling or sliding wear performance of pulleys, tire threads, bearings and camshafts materials. Hence, the contact area of the test specimen against the rotating counterface is variable, i.e., not constant. The longer the sliding distance with higher applied loads, the larger the contact area of the test specimen becomes. To balance the entire machine’s load lever arm, a counterweight balancer is placed at the end of the arm and adjusted such that the arm balances itself horizontally with respect to the pivot point. This is achieved when there is no load applied to the BOD machine. During the experiment, a load cell was directly fitted in the load lever of the machine to determine the frictional forces. Then, the frictional readings were obtained by incorporated a digital weight indicator, model DIBAL VD-310, directly to the load cell. Additionally, an infrared thermometer, model Extech 42580, was set and indicated at the interface of the test specimen to determine the interface temperature during the experiment. A weighing machine, Mettler Toledo MS105, was used to determine the weight loss for each specimen before and after the tribo test.

Dry sliding tests were carried out under different applied normal loads (5–30 N) at 6.72 km of sliding distance and different sliding velocities (Low: 1.7, Moderate: 2.83 m/s, High: 3.96 m/s). The specific wear rate (Ws) was computed using Equation (1), which is as follows:(1)$$W_{S}=\frac{\Delta V}{SDF_{N}}$$
where *W_S_* = specific wear rate in mm³/N·m; Δ*V* = volume difference according to changes of mass divided by density in mm³; *F_N_* =normal applied load in N; *SD* = sliding distance in m. The friction coefficient was computed by using Equation (2), which is as follows:(2)$$\mu =\frac{\mathrm{Measured}\;\mathrm{frictional}\;\mathrm{force}}{\mathrm{Normal}\;\mathrm{applied}\;\mathrm{load}}$$

More information on the BOR wear test rig is available at [2].

### 2.3. Examination of the Surface Roughness

A surface roughness instrument, the MarSurf Perthometer S2, was used to measure the roughness of the test specimens and stainless steel counterface surface subjected to the high applied normal load of 30 N and at high sliding speed of 3.96 m/s. These surfaces were measured before the test and after the test so that the difference in the surface roughness could be determined.

### 2.4. Examination of the Worn Surface Morphology

A metallurgical microscope, model MT8500, was utilized to capture the images of the worn surfaces’ morphology. The images were captured in a controlled environment with a humidity level of 90 ± 5% and room temperature of 32 ± 5 °C.

## 3. Results and Discussions

### 3.1. Wear Performance of BaFRE Composite

The specific wear rate vs. sliding distance for different fibre orientation (AP and P) subjected to different range of applied loads at different counterface sliding velocities are shown in Figure 4, Figure 5 and Figure 6. The wear properties of BaFRE composite were obtained in a controlled environment with humidity of 90 ± 5% at room temperature of 32 ± 5 °C. From the figure, it can be seen that the Ws is sensitive to the applied loads and sliding velocities of the counterface. In general, it can be observed that BaFRE composites for the AP-O exhibit better wear performance than the P-O.

From Figure 4, Figure 5 and Figure 6, it can be seen that the Ws had an exponential reducing trend with two different regions, namely the ‘running-in’ and ‘steady-state’ regions. The slope of the curves in the ‘running-in’ process was much greater at higher loads (20 N and 30 N) compared to the lower loads (5 N and 10 N), because higher loads will cause higher rate of weight loss. The wear rate of the BaFRE composite in AP-O took a shorter time to attain the steady-state transition, i.e., 5 km of sliding distance, while P-O demonstrated a longer time, i.e., 6 km. From the results, we can interpret that the BaFRE composite in AP-O has better wear performance than P-O due to the high shear resistance of the fibres in AP-O, which resisted the fibres detaching from the matrix. This had then reduced the material removal rate, thereby increasing the wear resistance in AP-O as compared to P-O.

The average specific wear rate as a function to different applied normal load and different counterface sliding velocities for AP-O, P-O and NE is presented in Figure 7. From the figure, it can be seen that the Ws of BaFRE composite in AP or P orientation is significantly more outstanding than in NE. Moreover, when the applied load increases, the wear rate for AP-O, P-O and NE tends to increase. Hence, the wear rate is influenced greatly by the increasing applied loads, i.e., higher material removal process. This will be further confirmed with the assistance of the morphology of the worn surfaces.

The wear performance for AP-O, P-O and NE under 30 N at different counterface sliding velocities is presented in Figure 8. From the figure, it is obvious that NE exhibits poor wear resistance compared with AP-O and P-O. This is mainly due to the thermos-mechanical loading incurred by NE due to the absence of the fibres. On the other hand, the corresponding relationship between the matrix and fibre had significantly enhanced the wear resistance of the composites where the matrix is responsible for consistently transmitting and distributing the applied load toward the fibres, while the fibres are shielding the matrix layer from being detached/fractured.

In a nutshell, the sequence of wear behaviour follows the order of AP-O > P-O >> NE, where NE experienced poor wear resistance. Thus, it could be said that reinforcing NE with banana fibres has the potential to enhance the composite’s wear behaviour.

### 3.2. Friction Performance of BaFRE Composite

The friction coefficient versus sliding distance for different fibre orientations (AP and P) subjected to a different range of applied loads at different counterface sliding velocities is presented in Appendix A: Figure A1, Figure A2 and Figure A3 respectively. A comparison of friction performance for AP-O, P-O and NE under different applied loads at different sliding velocities is presented in Figure 9. There is a decline in the friction coefficient with the increasing value of applied loads. This happened during dry sliding wear, i.e., when the applied load is increased, the temperature at the interface is also increased. The thermal stress in the specimen is generated by the high thermal gradients, thus weakening the fibre matrix bonding. In this way, fibres became loose, which resulted in fibre pull-out and detachment during sliding wear. These loose fragments could have acted as a third interface between the rubbing zone, which could have contributed to the friction coefficient reduction.

The friction coefficient of the composites under 30 N of applied load and 3.96 m/s of sliding velocity for AP and P orientation as well as neat epoxy is presented in Figure 10. One could interpret that, at low sliding velocity, the material removal process is slow and steady, which indirectly caused the test specimen’s contact to be more pronounced with the counterface. In other words, in AP-O samples, the resistance of the fibres came loose during the sliding, which was not the case for P-O since the fibres were in the direction of the sliding wear which eased the material’s removal process, i.e., a higher friction coefficient. In summary, the friction performance followed the order of NE >> P-O > AP-O. Hence, the current work suggests that by reinforcing epoxy with banana fibres and subjecting the contact condition in AP-O, there is a tendency to lower the friction coefficient as compared to P-O and NE.

### 3.3. Temperature Performance of the BaFRE Composite

The temperature profile graphs of BaFRE composite for AP-O and P-O as a function of sliding velocities and sliding distances are presented in Appendix B, Figure A4, Figure A5, Figure A6 and Figure A7 respectively. In general, when the specimens are subjected to a higher value of applied load, the temperature also rises. This is due to the thermo-mechanical loading that is the most important leading cause for dry sliding wear. Nevertheless, the contact mechanism of the AP-O composite exhibited low interface temperature due to a lower material removal process, since the fibres in AP-O had resisted coming loose during the sliding wear test. This is illustrated in Figure A7, which presents the temperature performance for AP and P fibre orientations as well as neat epoxy under different applied normal loads at various counterface sliding velocities. This implies that when banana fibres were oriented parallel to the sliding direction, there was a high rate of material removal; hence, the interface temperature tends to increase.

Figure 11 illustrates the temperature performance of AP and P fibre orientation as well as neat epoxy subjected to 30 N at different counterface sliding velocities. The figure concludes that banana fibres in AP orientations can reduce the interface temperature of the composite as compared to P orientations and neat epoxy. More justifications will be made with the evidence of the microscope images of the worn samples.

### 3.4. Surface Roughness of the Stainless Steel Counterface and BaFRE Composite

The surface roughness for the BaFRE composite and counterface for the different fibre orientation (AP and P) and neat epoxy subjected to 30 N and sliding velocity of 3.96 m/s is presented in Figure 12. It could be observed that the surface roughness increased for all the specimens after the tribo test; Figure 12a. The result indicates that the roughness for NE before and after the tribo test resulted in the lowest Ra values, followed by P-O and AP-O. Due to the absence of fibres in NE, the pure deformation of the sample was due to thermo-mechanical loading, which resulted in patches of melted epoxy on the worn surfaces that were obvious to the naked eye. This could be the reason for the low Ra values for the NE test samples.

On the other hand, the surface roughness of AP-O presented the highest Ra values while P-O experienced moderate Ra values. Figure 12b suggests that the counterface surface roughness was highly pronounced in AP-O as the fibres resisted the material removal process, which contributed to the high shear deformation in the rubbing zone.

The surface roughness profiles for different fibre orientation (AP and P) and neat epoxy under 30 N load at 3.96 m/s of sliding velocity are presented in Appendix C; Figure A8. It is observed that the slope of the surface roughness profile for AP-O is greater than P-O and NE. The value of composite surface roughness after the tribo test for AP-O is 1.122 µm, for P-O is 0.677 µm and for NE, on the other hand, is 0.221 µm. The counterface surface roughness (i.e. the Surface roughness of the counterface is provided in Appendix D; Figure A9 respectively) of AP-O after the tribo test exhibited the highest value because there was plenty of wear debris stuck on the stainless steel counterface. In a nutshell, it can be interpreted, for the current work, that P-O test samples demonstrated less damage to the counterface as compared to AP-O samples.

### 3.5. Examination of the Worn Surfaces’ Morphology

Microscopy images for the worn surfaces subjected to different applied normal loads at different sliding velocities for AP and P fibre orientations and neat epoxy are presented in Figure 13, Figure 14 and Figure 15. All specimens, after wear testing, normally show the signs of plastic deformation at the resinous region. This condition shows that the surface of the composite had very close contact with the counterface of the stainless-steel during sliding. The marks of micro and macro cracking (i.e., Figure 13f), debris formation and ploughing are observed on the worn surface of all the test specimens under different test conditions. In general, Figure 14 and Figure 15 revealed that the fibres were well adhered to the sliding direction of the counterface with less damage. This indicates that the BaFRE composite could withstand the load bearing capacity transferred to the banana fibres during the sliding wear test.

Nevertheless, for the absence of fibre in NE specimens, fractures at the resinous regions are noted on the worn samples; Figure 13g,h. This is because, due to the absence of the fibre, the brittle nature of epoxy resin increases, which will result in fracture, while surface roughness is affected by high friction. Interestingly, the rapid rise of interface temperature of neat epoxy subjected to higher applied loads generated severe plastic deformation on the worn surface because of thermo-mechanical loading. This causes the epoxy to melt on the worn samples which resulted in plastic deformation, as seen in Figure 13a–l, respectively.

In the case of AP-O, there is evidence of back film transfer on the worn samples; Figure 14f,g,k. This could explain the good wear behaviour of AP-O as compared to P-O and NE, since the back film transfer acts as a protection layer of shield to reduce the rate of the material removal process and further eliminate fibre detachment. At high sliding speed, debonding of fibre in the direction of sliding wear was noted; Figure 14f,l. However, the fibres were still in good shape, with no fibre tear noted on the worn surface; Figure 14e,k.

The wear modes of P-O subjected to different applied loads at different sliding velocities are presented in Figure 15. From the worn sample images, it is learned that there were signs of plastic deformation associated with rough surfaces and micro ploughing at low applied load; Figure 15a–c. When the load increased with increasing sliding speed, the wear was initiated with back film transfer, mild cracks in the resinous region and sign of detached fibres; Figure 15d–f. At 20 N of applied load and increasing sliding speeds, there was evidence of macro-ploughing and signs of trapped wear debris on the worn samples; Figure 15g–i. This could help justify the high values of Ws in P-O as compared to AP-O due to the ease of fibre detachment in P-O. When the load was 30 N and sliding speed increased, there was evidence of back film transfer and a high amount of wear debris scattered on the worn samples, Figure 15j–l. The evidence of wear debris suggests the high material removal process incurred, which could explain the high Ws and friction coefficient for P-O when compared to AP-O. Hence, the present work demonstrates that moderate wear of BaFRE composite is attainable when the composite is subjected to P-O.

All in all, it is understood that BaFRE composite revealed less damage than NE. Due to this, the possible suggestion of the proposed BaFRE composite can be made related to non-structural applications [20,21], since many attempts have been made to use natural fibre composites in place of pure epoxy resin. In addition to this, [22,23,24,25] reported that a good number of automotive components previously made with NE are now being manufactured using low carbon base content composites. Eberle and Franze [26] revealed that automotive giants, such as Daimler Chrysler and Mercedes Benz, are continuously producing low weight vehicles using natural fibre composites, since for every 1 kg of weight reduction of an automobile vehicle, about 5 to 8 L of gasoline can be saved. Secondly, Bhushan [27] and Cirino et al. [28] had suggested that natural fibre composites can be used as bearing and sliding materials subjected to tribological loading conditions due to their low friction conditions, high wear resistance and easy process ability properties. For the current work, it is proposed that the BaFRE composite could be used in the application of automotive door guide rails. Besides the claims made by [27,28], it is learned that many vehicles have adopted the use of thermoset plastic resins which are non-biodegradable and high in density as compared to natural fibre composites. Figure 16 is a typical example of a vehicle model Mazda CX5 where the door guide rail linkage is made of thermo plastic resin [29]. A good substitute will be BaFRE composite in AP-O, since the current work demonstrated that AP-O has good tribological properties as compared to P-O and NE. Though BaFRE composite offers several benefits as compared to NE and other synthetic composites, several major technical considerations must be addressed before the engineering, scientific and commercial communities gain the confidence to enable wide-scale acceptance, particularly in exterior parts, where a Class A surface finish is required. To name but a few, these challenges include in-depth investigation on the homogenization of the fibre properties and a full understanding of the degree of polymerization and crystallization, adhesion between the fibre and matrix, moisture repellence and flame retardant properties. Hence, the aforementioned challenges may open new research pathways to the scientific community in the future.

## 4. Conclusions

After conducting this research, the following conclusions are proposed:(i)The wear performance followed the order of AP-O > P-O > NE.(ii)AP-O had brilliant wear resistance at lower counterface sliding velocity under a lower applied load.(iii)The Ws was improved by about 6.37% and 40.47% for P-O and AP-O at 1.7 m/s of sliding velocity as compared to NE.(iv)The Ws was improved by about 4.99% and 29.32% for P-O and AP-O at 2.83 m/s of sliding velocity as compared to NE.(v)The Ws was improved by about 2.74% and 13.44% for P-O and AP-O at 3.96 m/s of sliding velocity as compared to NE.(vi)There is no correlation between wear performance and friction performance of the composites.(vii)The nature of friction behaviour is determined by the condition of fibres and formation of back film transfer at the interface.(viii)The friction coefficient was decreased by about 2.99% and 53% for P-O and AP-O at 1.7 m/s of sliding velocity as compared to NE.(ix)The friction coefficient was decreased by about 23.4% and 43.8% for P-O and AP-O at 2.83 m/s of sliding velocity as compared to NE.(x)The friction coefficient was decreased by about 19.4% and 48.9% for P-O and AP-O at 3.96 m/s of sliding velocity as compared to NE.(xi)The wear mechanisms that were incurred in different fibre orientations are dissimilar, such as: AP-O: back film transfer and debonding; P-O: deteriorated and detached fibre; NE: fracture and rough surface.(xii)AP orientation of the composites exhibited outstanding friction and wear performance compared to P orientation and NE.

(Note: For items iii, iv, v, viii, ix and x, the percentage values were computed from Figure 7 and Figure 9, respectively, and compared to the base material, which is NE.)

## Figures and Tables

**Figure 1 polymers-14-03700-f001:**
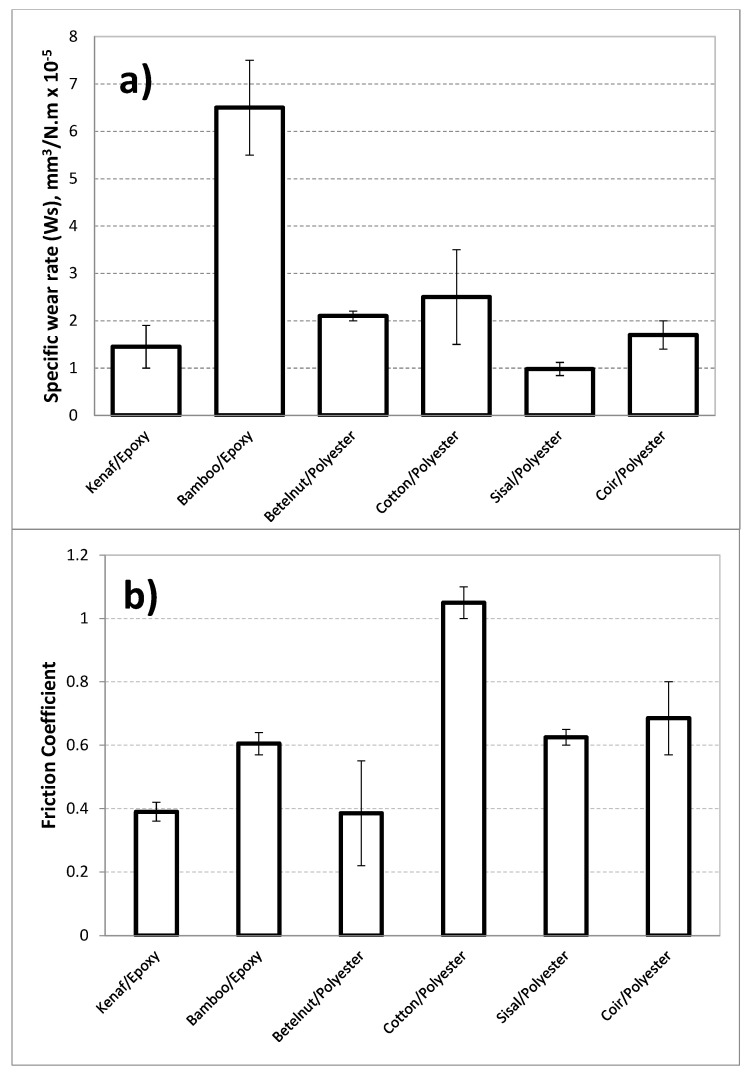
Specific wear rate (**a**) and friction coefficient (**b**) of some polymeric composites under dry contact conditions [16].

**Figure 2 polymers-14-03700-f002:**
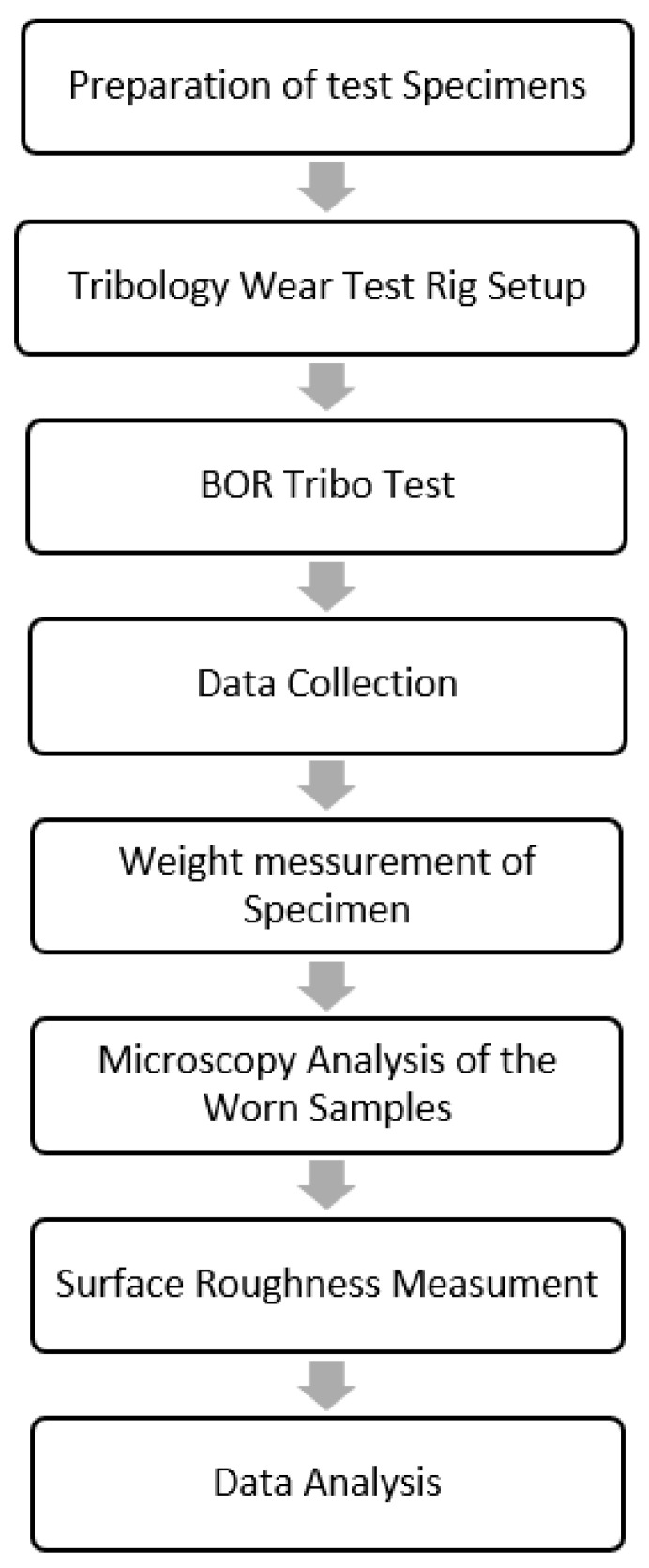
Flow chart showing sequence of works on tribological performance of BaFRE composite.

**Figure 3 polymers-14-03700-f003:**
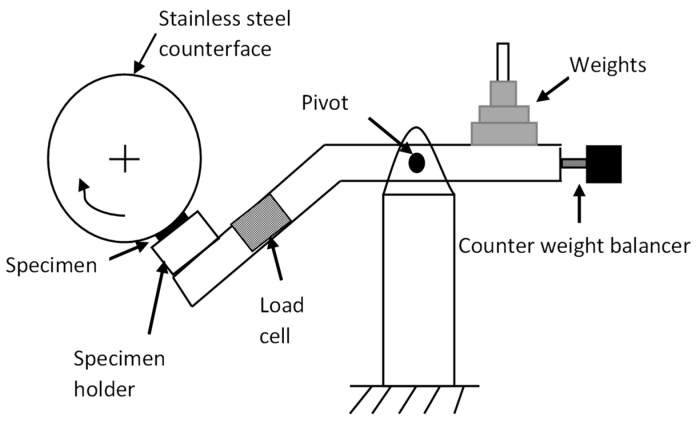
Schematic view of a BOR tribo test machine.

**Figure 4 polymers-14-03700-f004:**
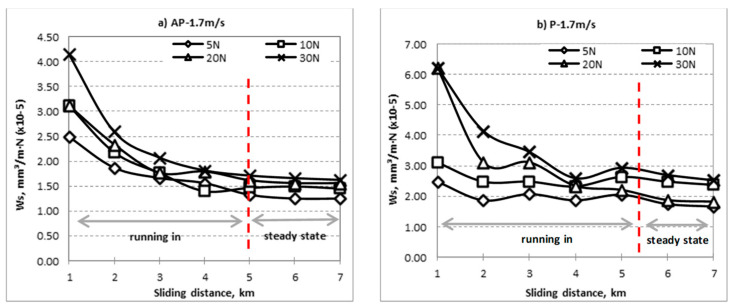
Specific wear rate vs. sliding distance for different fibre orientation (AP and P) subjected to difference of applied normal load at 1.7 m/s of sliding velocity.

**Figure 5 polymers-14-03700-f005:**
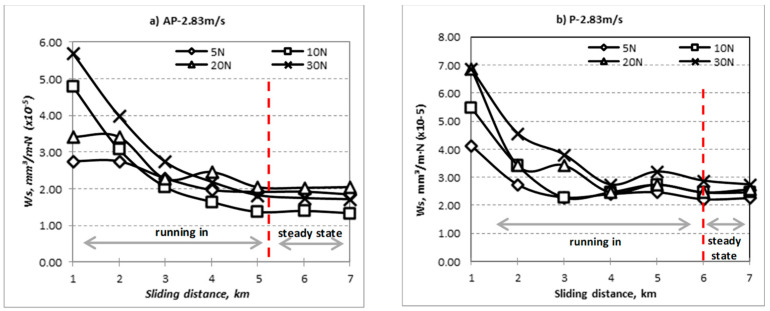
Specific wear rate vs. sliding distance for different fibre orientation (AP and P) subjected to difference of applied normal load at 2.83 m/s of sliding velocity.

**Figure 6 polymers-14-03700-f006:**
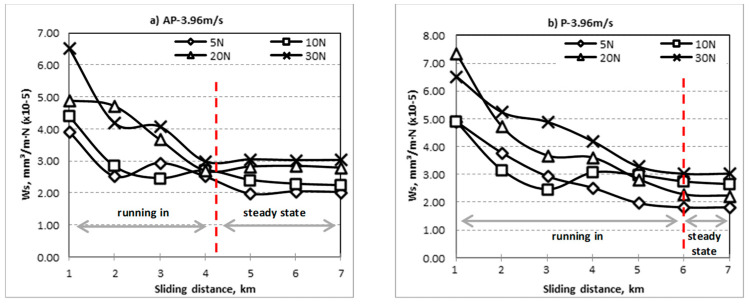
Specific wear rate vs. sliding distance for different fibre orientation (AP and P) subjected to difference of applied normal load at 3.96 m/s of sliding velocity.

**Figure 7 polymers-14-03700-f007:**
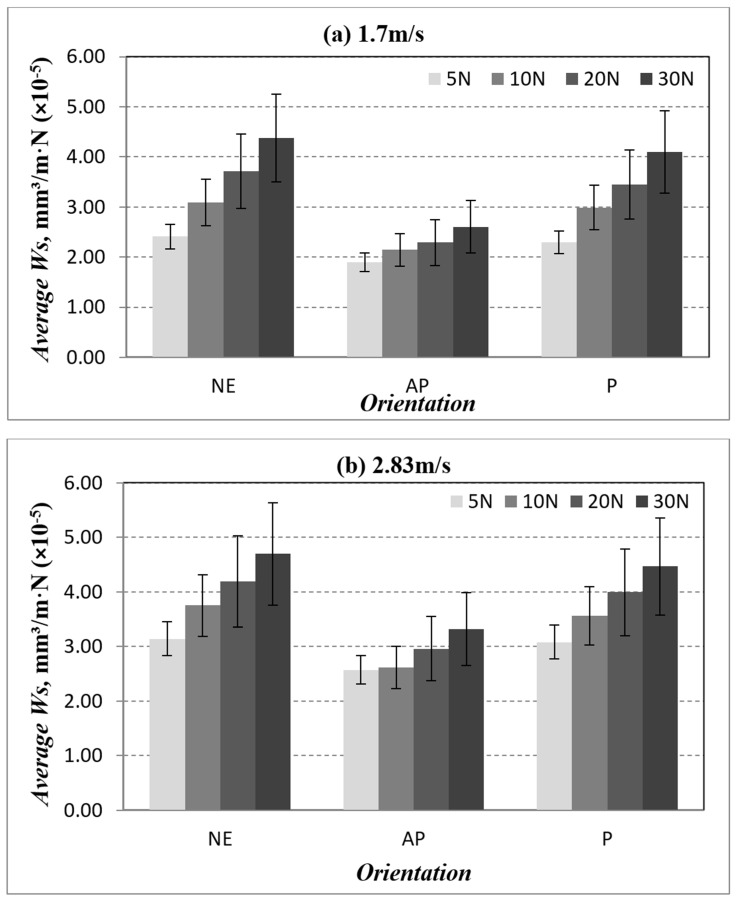
Average specific wear rate for different fibre orientation (AP and P) and neat epoxy subjected to different of applied normal load at different of sliding velocities.

**Figure 8 polymers-14-03700-f008:**
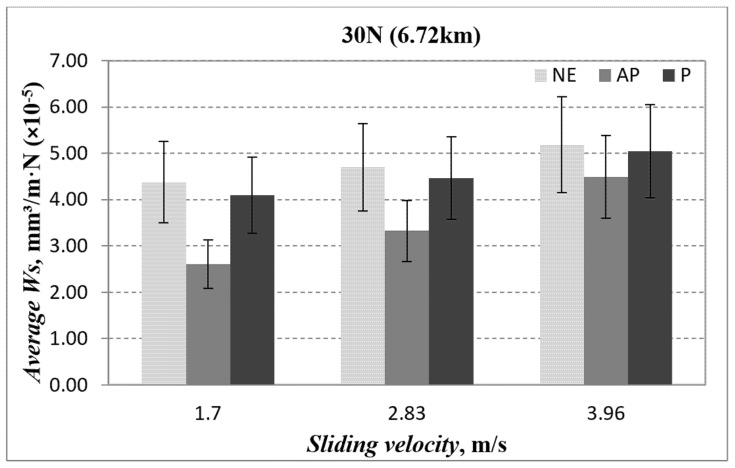
Average specific wear rate vs. sliding velocity for different fibre orientation (AP and P) and neat epoxy subjected to an applied normal load of 30 N at different of sliding velocities.

**Figure 9 polymers-14-03700-f009:**
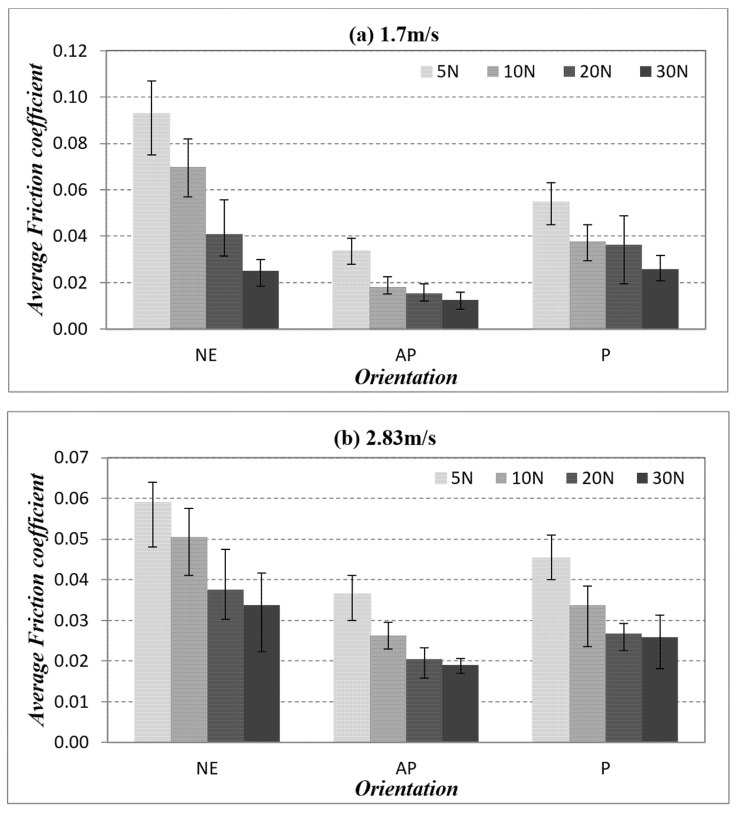
Friction coefficient for different fibre orientation (AP and P) and neat epoxy subjected to different of applied normal load at different of sliding velocities.

**Figure 10 polymers-14-03700-f010:**
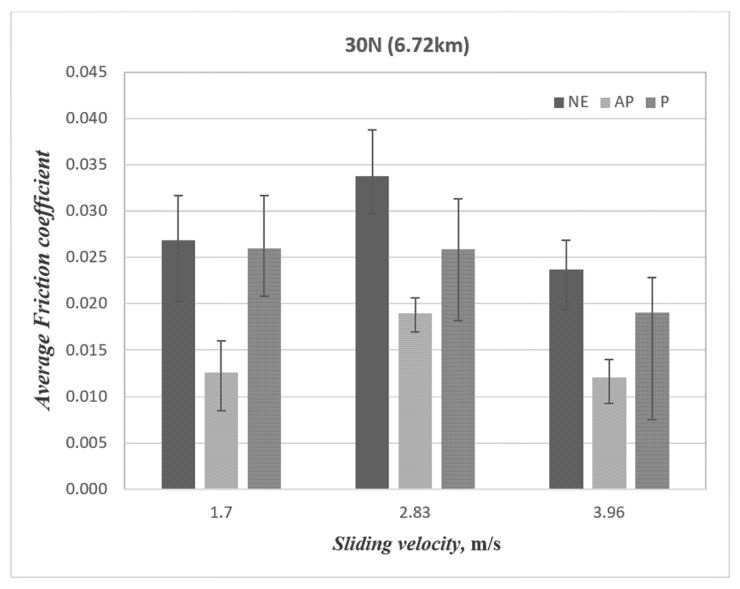
Friction coefficient vs. sliding velocity for different fibre orientation (AP and P) and neat epoxy subjected to 30 N of applied load at different of sliding velocities.

**Figure 11 polymers-14-03700-f011:**
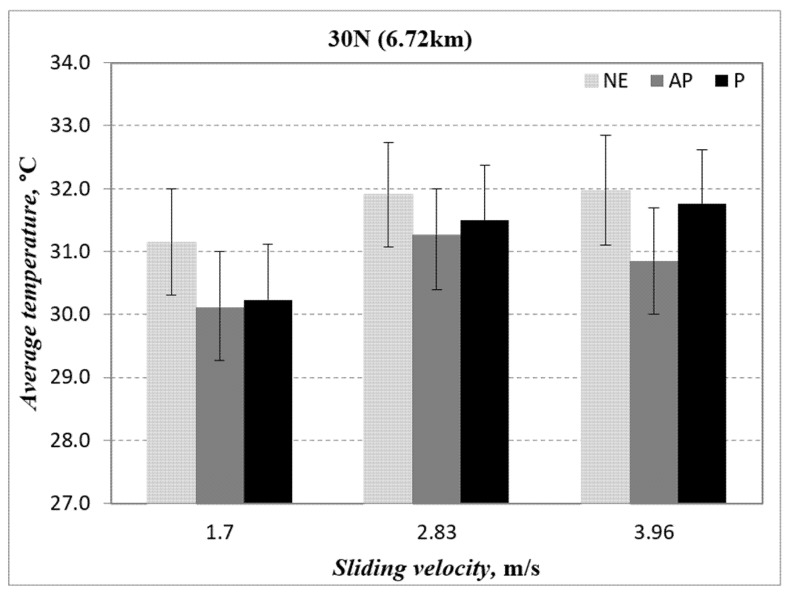
Temperature vs. sliding velocity for two different fibre orientations (AP and P) and neat epoxy subjected to an applied normal load of 30 N at 3.96 m/s of counterface sliding velocities.

**Figure 12 polymers-14-03700-f012:**
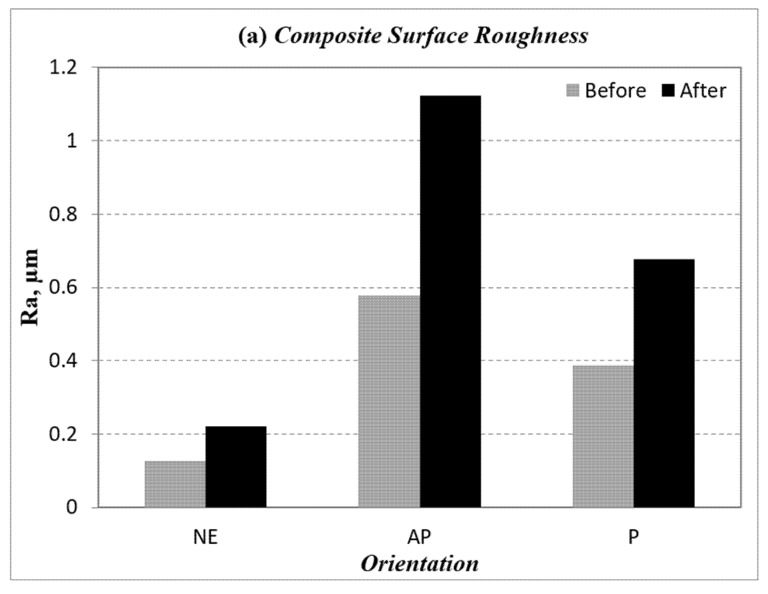
Composite and counterface surface roughness for different fibre orientation (AP and P) and neat epoxy.

**Figure 13 polymers-14-03700-f013:**
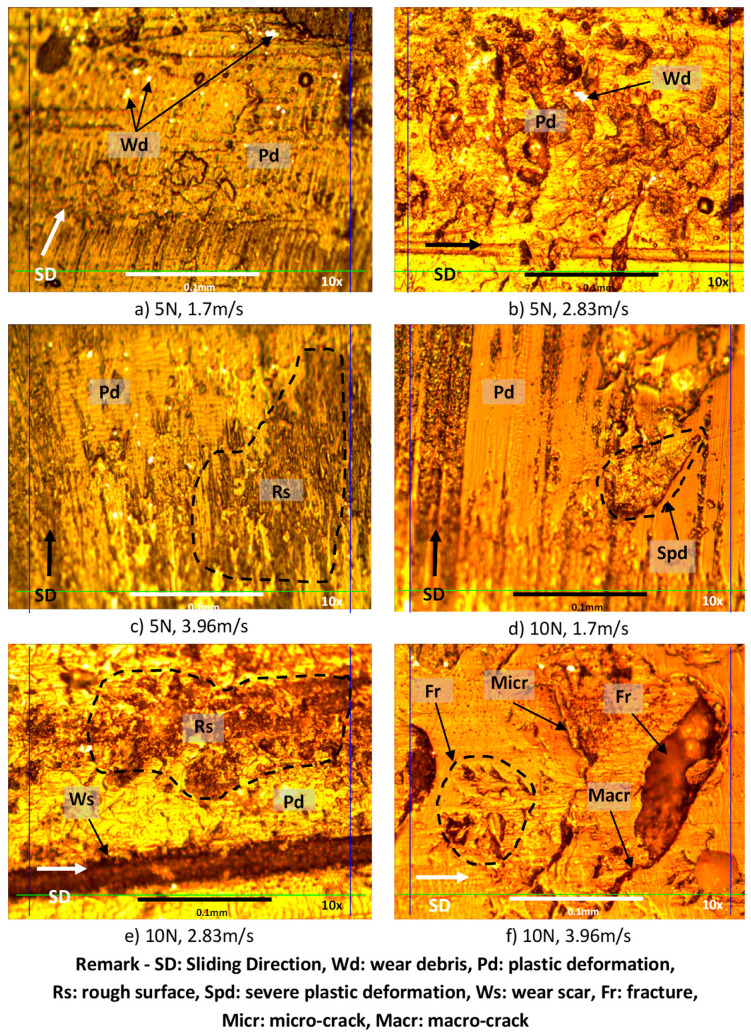
Photo micrographs of neat epoxy tested under different applied normal loads at different counterface sliding velocities.

**Figure 14 polymers-14-03700-f014:**
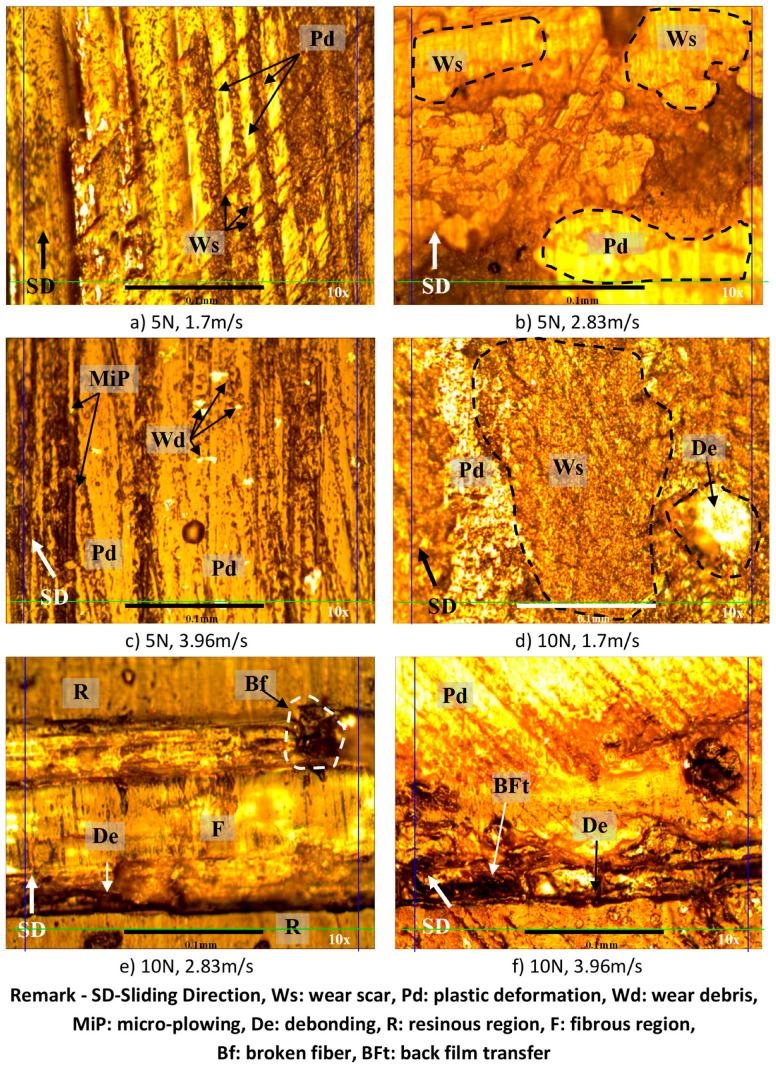
Photo micrographs of BAFRE composite tested in AP-O under different applied normal loads at different counterface sliding velocities.

**Figure 15 polymers-14-03700-f015:**
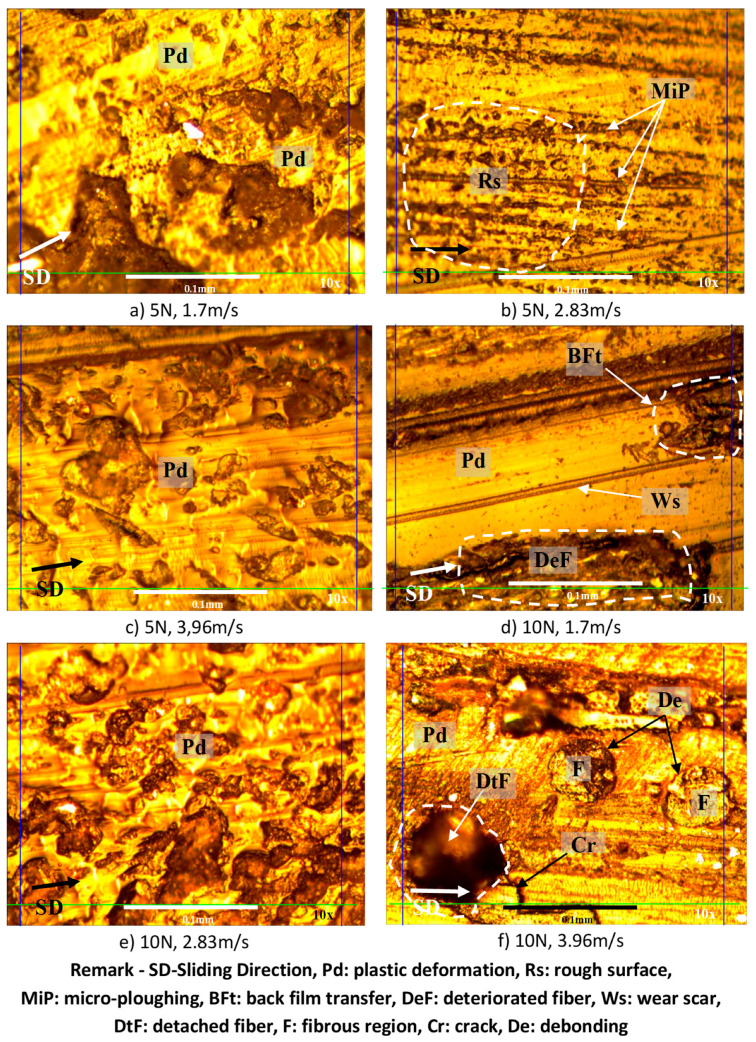
Photo micrographs of BAFRE composite tested in P-O under different applied normal loads at different counterface sliding velocities.

**Figure 16 polymers-14-03700-f016:**
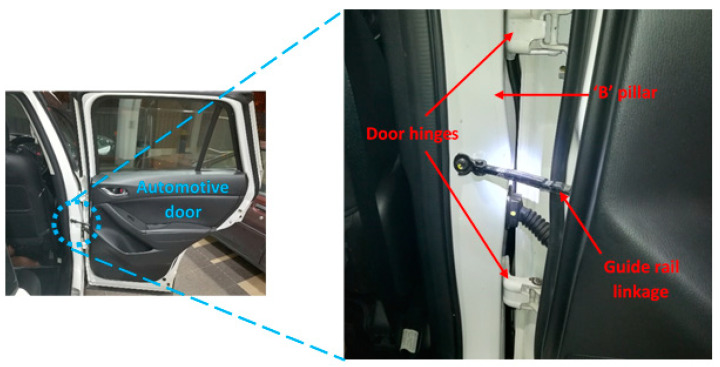
A typical linkage working operation of an automotive vehicle model: Mazda CX-5 [29].

## Data Availability

Not applicable.

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
