# Peer review of "Adhesive Wear and Frictional Performance of Banana Fibre Reinforced Epoxy (BaFRE) Composite"

_polymers, 2022, doi:10.3390/polym14173700_

Round 1
Reviewer 1 Report
Reviewer’s comment:
This manuscript deals with the tribological performance of banana fibre reinforce epoxy (BaFRE) composite at dry contact conditions, which are generally used in in many applications. In all, the paper falls within the scope of this journal and is written in a clear way and most of the claims are supported by data and figures. However, reporting a list of experimental results is not enough. Further, author should consider the relationship between the mechanism tribological performance and the materials.
Thus, I indeed think the paper can be accepted after a proper revise.
(1) P4, “Materials” which contains reagents, manufacturers and even chemical structure should give a detailed description that other scholars can repeat the plating process.
(2) P7, Figure 5 and Figure 6 have no sign of 20 N and 30 N.
(3) Line 260-262 of P12, “there are absence of fibres in the neat epoxy that make its surface smooth and flat” should be characterized by AFM.
(4) Reporting a list of experimental results is not encouraged in academic paper. Thus, authors should present a deep current discussion.
In addition, there are some grammatical errors, misuse of some terms, and many sentences do not be read well more or less. For example:
1. LINE 348, “see Figure 16”
2. Line 360, a few important points can be drawn which is
Reviewer 2 Report
The author has investigated the tribological performance of banana fibre reinforced epoxy (BaFRE) composite at dry contact conditions. Different applied normal loads (5N - 30N) subjected to a stainless steel counterface at different sliding speeds (1.7 - 3.96m/s) and fix sliding distances of 6.72 km were set as the experimental parameters. The author reported that there are improvements in terms of wear and friction performance of the composite when banana fibres are used as a reinforced element in epoxy resin. The work seems good, but the author must reply to the following query response and corresponding changes in the manuscript is required.
1. Author needs to revise the whole manuscript carefully to eliminate mistakes in the manuscript.
2. Page 3 line 80 “it was noted that there is no work reported on banana fibre reinforced epoxy composite for tribological applications” but there are many works that have been reported in this field how can the author claim this.
3. What is the G137-95 standard? Kindly explain with citations.
4. For Examination of the worn surface morphology SEM should be used instead of an optical microscope for better clarity. Kindly provide the reason behind optical microscope uses with justification.
5. In Figures 4 and 6 only two forces are visible but there should be four forces as in Figure 5.
6. Remarks must with written in-text “SD-Sliding Direction (Wd: wear debris, Pd: plastic deformation, Rs: rough surface, Spd: severe plastic deformation, Ws: wear scar, Fr: fracture, Micr: micro-crack, Macr: macro-crack)”. The levelling of optical images is incomplete. Rs as R, Spd as S in levelled in Image. Kindly revised levelling.
7. The results discussion needs to be revised with clear data of the results obtained.
8. Page 20-21 lines 333-358 are just like literature why it is placed here in discussions.
9. The conclusion needs to be revised with the reason behind any results like why percentage improvement is obtained in the case of P-O and AP-O.
Reviewer 3 Report
Look comments

Round 2
Reviewer 2 Report
Authors have responded to almost all queries satisfactorily.
Reviewer 3 Report
It can be enough for considerations